# Non-Conventional Time Domain (TD)-NMR Approaches for Food Quality: Case of Gelatin-Based Candies as a Model Food

**DOI:** 10.3390/molecules27196745

**Published:** 2022-10-10

**Authors:** Sirvan Sultan Uguz, Baris Ozel, Leonid Grunin, Emin Burcin Ozvural, Mecit H. Oztop

**Affiliations:** 1Department of Food Engineering, Middle East Technical University, Ankara 06800, Turkey; 2Resonance Systems GmbH, 73230 Kirchheim unter Teck, Germany; 3Department of Food Engineering, Ahi Evran University, Kirsehir 40100, Turkey; 4Department of Food Engineering, Çankırı Karatekin University, Çankırı 18200, Turkey

**Keywords:** soft candy, gelatin, confectionery, TD-NMR, crystallinity, longitudinal relaxation time, spin diffusion, Magic Sandwich Echo, Solid Echo, second moment

## Abstract

The TD-NMR technique mostly involves the use of T_1_ (spin-lattice) and T_2_ (spin-spin) relaxation times to explain the changes occurring in food systems. However, these relaxation times are affected by many factors and might not always be the best indicators to work with in food-related TD-NMR studies. In this study, the non-conventional TD-NMR approaches of Solid Echo (SE)/Magic Sandwich Echo (MSE) and Spin Diffusion in food systems were used for the first time. Soft confectionary gelatin gels were formulated and conventional (T_1_) and non-conventional (SE, MSE and Spin Diffusion) TD-NMR experiments were performed. Corn syrups with different glucose/fructose compositions were used to prepare the soft candies. Hardness, °Brix (°Bx), and water activity (a_w_) measurements were also conducted complementary to NMR experiments. Relaxation times changed (*p* < 0.05) with respect to syrup type with no obvious trend. SE/MSE experiments were performed to calculate the crystallinity of the samples. Samples prepared with fructose had the lowest crystallinity values (*p* < 0.05). Spin Diffusion experiments were performed by using Goldman–Shen pulse sequence and the interface thickness (*d*) was calculated. Interface thickness values showed a wide range of variation (*p* < 0.05). Results showed that non-conventional NMR approaches had high potential to be utilized in food systems for quality control purposes.

## 1. Introduction

Gelatin-based candies are popular confectionery products composed of gelatin as the gelling agent, sucrose, glucose (corn) syrup, coloring, acid, and flavoring agents [1]. Sugar is added to these jellies not only for the textural properties but also to prevent microbial growth [2].

Time Domain-NMR (TD-NMR) techniques have been used for quality control purposes in foods for many years [3,4,5]. In TD-NMR studies, most of the time, relaxation times of the samples are measured to get insight into the quality attribute of interest. Spin-lattice (longitudinal) relaxation time (T_1_) is the time required for the spins to realign along the longitudinal z axis. Spin-spin relaxation (T_2_) or transverse relaxation time is defined as the time that determines the rate at which the x-y component of the magnetization (M_xy_) decays [6].

In addition to the relaxation time measurements, there are other TD-NMR techniques which can be used to extract more information on the samples. Solid Echo (SE) is one such pulse sequence that allows partial refocusing of the solid fraction and attains the crystallinity of a sample by back extrapolation over a series of experiments carrying echo delays. As the second pulse is applied out of phase, it allows for signal to be obtained from the initial part of the FID, thus resolving the dead time issue [7]. While SE mainly refocuses dipole interactions within a pair of spins, the initial part of FID can also be refocused by using Magic Sandwich Echo (MSE) sequence with correct parameters and a phase cycling routine [8]. It is a modified form of SE which prevents the dead time problem by refocusing the initial part of the free induction decay (FID) as well. MSE has proven to be a more robust method to investigate polymer mobility, and compared to SE, it allows a better refocusing of multi spin dipolar interactions [9]. However, SE sequence is a much shorter and faster sequence compared to MSE which requires at least four phase cycling steps. In this study, MSE has been used for the characterization of soft candies for the first time. 

Another non-conventional TD-NMR approach is spin diffusion which is observed when there is a gradient of magnetization between phases with different mobility. The magnetization leakage itself is observed by the reduction of the “long” component of FID and simultaneous increase of the “short” component contribution into the overall signal during the increase of the spin diffusion time [10]. Spin diffusion can be used for characterization of the interface between amorphous and rigid crystalline phases. The crystalline size can be investigated as a function of crystallization time with spin diffusion in addition to the total crystalline content [11]. Spin diffusion has also been conducted for evaluation of the linear sizes of the surface of cellulose crystallites [10].

In this study, gelatin candies prepared with different corn syrup types were selected as model food and effects of syrup types-concentration on candies were examined by using different TD-NMR techniques. Five different corn syrups with different glucose/fructose ratios were used. The amounts of these syrups were varied and replaced by sucrose alternatively. T_1_ measurements were conducted to support the information obtained from SE/MSE and Spin Diffusion experiments. To complement the information obtained by NMR experiments, water activity (a_w_), total soluble solids (°Bx), and hardness values of the soft candies were also measured since these parameters of the candies were also affected by water–syrup interactions. The results of this study could provide input to use SE, MSE, and spin diffusion in different food systems.

## 2. Results and Discussion

### 2.1. Brix

Brix values of the corn syrups used in the study changed in the range of 79–83 °Bx. While SCG60 and SCG40 syrups had the highest Brix values, SMF42 had the lowest one. Corn syrup type was found to affect the Brix values of the candy samples significantly (*p* < 0.05). However, the amount of corn syrup used (30% and 60%) did not have a significant effect (*p* > 0.05) on the Brix values of the same candies (Figure 1). Candy samples prepared with SCG60 syrup had the highest Brix values (*p* < 0.05). One of the reasons for the high Brix of SCG60 candies could be the high Brix of the SCG60 syrup that was used in the candy formulation. This syrup contains the highest amount of glucose (60.5%). Presence of a high amount of glucose in candy formulations is associated with higher Brix values [12]. Consequently, SCG60-60% candies attained the highest glucose concentration (34.2 g/100 g mix), thereby showing the highest Brix (~84 °Bx) among all candies. The rest of the candy samples demonstrated similar Brix values despite the differences in their formulations. In fact, higher Brix values could be expected for the samples prepared with 60% corn syrup since these samples contain higher amounts of total glucose and fructose concentration. Fructose is the most water soluble sugar [13] and contributes greatly to total soluble solid content of confectionery formulations. However, this was not the case as demonstrated. Presence of 30% sucrose in the samples prepared with 30% corn syrup somehow balanced the Brix values with those prepared with 60% corn syrup. One of the reasons could be the introduction of higher amount of higher molecular weight (MW) sugars such as maltose and some oligosaccharides into the candy formulations at 60% corn syrup concentration. The longer chains of such higher MW sugars may have experienced a lower solubility and may also have affected the solubility of glucose and fructose in the system [14].

### 2.2. Water Activity

Water activity experiments were also performed for the candies. Except for the SCG40 samples prepared with 60% corn syrup (SCG40-60%), a_w_ of all samples were in the expected range (≤0.70) as shown in Figure 2. Statistical analysis indicated that both the type and amount of the syrup used in the candy formulations significantly (*p* < 0.05) affected the a_w_ of the samples. Figure 2 demonstrates that the SCG40-60% candy had the highest (*p* < 0.05) a_w_ (~0.85). Since concentration of glucose is higher in SCG60-60% candies, it is reasonable for these samples to have lower a_w_ with respect to those of SCG40-60% candies. More glucose molecules induce more H-bonding, and thus decrease a_w_. However, the highest a_w_ of the SCG40-60% is still interesting. The lack of fructose which is the most soluble sugar that would induce lower a_w_ in candies could be one of the reasons for this phenomenon [13]. All samples other than SCG40-60% attained lower a_w_ values between 0.6 and 0.7 (Figure 2).

### 2.3. Hardness

In addition to °Brix and a_w_ experiments, hardness values of the candies were also measured (Figure 3). Corn syrup type and concentration were again significant on the hardness of the candies (*p* < 0.05). SCG60-60% samples had the highest hardness (*p* < 0.05) which was in agreement with their high °Brix and low a_w_ values [15]. High Brix and low a_w_ indicate that the water is entrapped within the candy matrix efficiently and homogeneously [16]. In contrast, both of the SCG40 candies, especially the one prepared with 60% corn syrup, were the softest samples. SCG40-60% showed a very soft texture that was below 10 N as shown in Figure 3. This is actually compatible with the substantially high a_w_ of these samples. At such a high a_w_ level, it is apparent that the continuous phase of the soft candies was not homogeneous due to the poor affinity of water with the sugar constituents of the samples [17]. Therefore, these candies could not preserve their shape and collapsed under even low forces. The remaining samples (SMF42, SBF10, and SHFSLF20) achieved statistically similar hardness values regardless of the corn syrup concentration used.

### 2.4. Longitudinal Relaxation Times

T_1_ data were fitted to a mono-exponential model. Both syrup type and concentration affected T_1_ significantly (*p* < 0.05). T_1_ values changed in the range of 35–65 ms and the longest T_1_ was observed for the SCG40-60% samples whereas the shortest T_1_ was observed for the SMF42-30% samples (Figure 4). When the syrup type was considered as the main factor, it was observed that SMF42 and SHFSLF20 samples had the shortest mean T_1_ (average of 30% and 60% syrup samples) values (*p* < 0.05). T_1_ is shortest when the molecular tumbling rate (also known as the correlation time, τ_c_) is approximately equal to the Larmor frequency [6]. This tumbling rate also depends significantly on the viscosity/hardness of the sample. At lower viscosities/soft textures and higher free water populations, higher T_1_ values are expected for the same samples [18]. This could be the case for SCG40-60% candies whose a_w_ reached the highest level. These samples had a higher proportion of free water molecules contributing to high a_w_ and as a result, SCG40-60% samples attained the longest T_1_ (*p* < 0.05).

Another factor that may have contributed to the long T_1_ of SCG40-60% candies could be the crystallinity characteristics of these samples [19]. In theory, molecules tumbling faster or slower than in ω_0_τ = 1 condition are less efficient at spin-lattice relaxation and have longer T_1_. Therefore, free water or crystals have longer longitudinal relaxation times due to the wide range of tumbling rates [20] and most of the molecules in this state (free water and/or crystals) are inefficient at spin-lattice relaxation. The possible effects of crystallinity will be discussed in the upcoming section.

### 2.5. Second Moment

SE and MSE sequences were used to calculate the second moment (M_2_) which has been shown to be an important parameter for estimating the crystallinity of different samples [21]. The approach described by Grunin et al. (2019) was used by employing the module in Relax 8 software (Resonance Systems GmBH, Kirchheim, Germany) [8]. M_2_ results of the samples for SE and MSE are given in Table 1.

As expected, M_2_ values calculated from MSE and SE were highly correlated (R = 0.93, *p* < 0.05). This also confirmed the accuracy of the analysis. M_2_ is affected by the proton mobility within a crystal lattice, thereby providing valuable information on the molecular dynamics in the crystalline state [21]. The lower proton mobility in a solid/crystalline state stimulates and enhances the dipolar interactions between the protons, resulting in higher M_2_ [22]. Thus, higher M_2_ values could be associated with higher degrees of crystallinity in a system [8]. It should also be noted that the M_2_ values obtained in this study are just relative values. For the exact quantification of the crystallinity, a calibration curve should be prepared as in the study of Grunin et al. (2019) [8]. Crystallinity values could also be calculated from another conventional method such as X-ray diffraction (XRD). However, in practice, it is not feasible to do this for different samples. Therefore, in this study, rather than calculating the exact crystallinity values, a comparison between the samples was performed to show whether samples prepared from different corn syrup samples had different crystallinity. In this way, effects of corn syrup concentration and presence or absence of sucrose in the formulations were also evaluated.

According to Table 1, SCG40-60% samples achieved one of the highest crystallinity values whereas SMF42-60% samples had the lowest ones (*p* < 0.05). Glucose is one of the factors that may have contributed to the higher M_2_ values. Glucose is known to have lower solubility than fructose and its crystallization tendency is quite high. For instance, in honey, which is a syrup composed of fructose and glucose, crystals are formed from glucose [23]. In that regard, the samples prepared with the syrups having high glucose content are expected to have a higher amount of crystals and this was reflected in both SE and MSE results as seen in Table 1. Additionally, SCG60-60% candies with the highest amount of glucose (34.2 g/100 g mix), also possessed high M_2_ similar to SCG40-60%. Another factor that is effective on M_2_ could be the degree of interactions taking place between the glucose molecules. As previously mentioned, SCG40-60% samples showed very high M_2_ with glucose amount of 22 g/100 g mix in their formulation. Although this is not the highest glucose amount in the candy formulations studied, the nature of the interactions between the glucose molecules may have played a crucial role on the M_2_ results. This candy formulation had the highest a_w_ as discussed in the previous sections. This suggests that the majority of the water molecules were in a free state. Thus, glucose–glucose interactions were favored over glucose–water interactions in this case [6]. This probably enabled the formation of a compact and highly ordered glucose crystal lattice that contributed to the crystallinity level of these samples [24]. Fructose is also an important factor affecting the M_2_ results. Although SMF42-60% candies also had a considerable amount of glucose (28.8 g/100 g mix) in their structures, they possessed the lowest M_2_ (*p* < 0.05). The reason is the presence of high amount of fructose (24 g/100 g mix) in these samples. Fructose molecules have a low tendency for crystallization and may act as doctoring agents via interacting with the glucose molecules [25]. Consequently, formation of a glucose crystal lattice was restricted in SMF42-60% samples [26]. SBF10-60% samples also experienced lower M_2_ than the maximum value due to their high fructose content. Finally, presence of sucrose should be considered for the M_2_ results. When Table 1 was reviewed, one could observe that all samples containing 30% sucrose had statistically the same or higher M_2_ with respect to their counterparts with no sucrose. Although sucrose-containing samples have lower amounts of glucose in their formulations, sucrose crystallization compensated the M_2_ of these samples since crystal lattice formation by sucrose molecules is also a common phenomenon observed in confectionery products [27]. Another point that needs attention is that the candies visually did not suffer from sugar crystals. Nevertheless, the power of non-conventional TD-NMR provided the ability to detect even low numbers of crystals within the candies. Additionally, M_2_ values of sucrose crystals and dried gelatin samples were also determined. Sucrose crystals had M_2_ of 14.63 and 12.34 based on SE and MSE measurements, respectively. This high sucrose M_2_ indicated that all analyzed samples reached partial crystalline regions. On the other hand, dried gelatins attained M_2_ of 10.19 and 9.24 based on SE and MSE measurements, respectively. This suggests that the dried gelatin fractions within the samples also contributed somewhat to the overall crystallinity of the analyzed samples.

### 2.6. Spin Diffusion

Spin diffusion occurs when there is a gradient of magnetization between phases with different mobility. The polarization transfer is the leakage of magnetization from ^1^H nuclei of soft protons to the crystalline and amorphous regions. The leakage itself is observed by the reduction of the “long” component of FID and simultaneous increase of the “short” component contribution into overall signal during the increase of the spin diffusion time [10]. In our case, long component refers to the water that is “trapped in the gel” whereas short component is the solid portion and in the case of a soft candy, it is the crystal sugars that could potentially form. Glassy polymer network could also contribute to the solid portion. Spin diffusion leads to communication among spins at various sites in a solid. It has been shown that such communication could provide useful information about the spatial inhomogeneity in the resonance frequencies and relaxation times. The rate of the spin–flip communication is directly proportional to the local dipolar field [28]. Thus, we hypothesized that this behavior could help us understand the differences in different candy formulations prepared at different formulations. The challenge of spin diffusion experiments is the data analysis conducted afterwards. What we get from the spin diffusion is a parameter that is related with the domain sizes which can be used as a characterization parameter for different candies [29,30]. It is hard to visualize it physically in a soft candy structure since the matrix is complex. Thus, domain size may not have a real physical meaning. However, it could be used to obtain a quantitative parameter that will let researchers to make comparison between different samples. And it was good to see that magnetization transfer literally occurred in the candy samples.

In order to calculate the so called ‘domain size’ parameter, M_2_ (calculated from MSE experiments) was used first to calculate the effective spin diffusion coefficient (D_sd_). r^2^ (mean square distance between spins) value was estimated within a range of 0.22–0.25 nm similar to cellulose since average distance was nearly similar in all saccharides [28].

Since spin diffusion is guided by dipole–dipole spin coupling, and the second moment of an NMR line is also the consequence of the dipolar constant [31,32], the following equation can be utilized to derive the spin–diffusion coefficient from results of MSE measurements:(1)$$D_{\mathrm{sd}}=\frac{\sqrt{\pi }}{6}r^{2}\sqrt{M_{2}}$$

The transfer thickness (*d*) (the domain size parameter) of the interface layer where magnetization transfer occurred was found according to well known “initial rate approximation” approach that is well described in [11,28,33,34];
(2)$$d=\frac{2{\beta t}^{0.5}}{\sqrt{\pi }}\sqrt{D_{\mathrm{sd}}}$$
assuming that the magnetization transfer was carried out in one direction (β = 1).

t^0.5^ was calculated by conducting a Goldman–Shen experiment (Figure 5) [35]. Linear fitting was performed by using Origin software. Results of the ‘*d*’ values are given in Table 2.

Both syrup type and concentration were found to be significant on the ‘*d*’ values (*p* < 0.05). Higher syrup concentrations resulted in larger thickness values as shown in Table 2. Candies prepared with glucose syrups of SCG40 and SCG60 had similar values at each syrup concentration whereas the candy samples prepared with fructose syrup differed significantly from each other (*p* < 0.05) and showed high variations. It was quite interesting to see that as fructose content of the syrup increased in the samples prepared with a fructose syrup, interface thickness also increased. There was a monotonic increasing trend of the ‘*d*’ values in the fructose soft candies with respect to fructose concentration of the syrup. The extent of this increase was even greater in the samples containing 60% fructose syrup (Table 2). The transfer thickness (*d*) is defined as the thickness of the interface layer which is between the bounded water and crystalline phase [36]. Thus, the presence of fructose increased this thickness and led to lower crystallinity which was also confirmed by the lower crystallinity values of the fructose-containing samples obtained in SE/MSE experiments (Table 1). On the other hand, soft candies prepared with SCG syrups demonstrated statistically the lowest ‘*d*’ values (*p* < 0.05) which was also consistent with their high M_2_ results. These results showed that if the water binding is high, interface layer thickness increases and thus crystallinity decreases. The different interface thicknesses imposed by the use of different syrups indicated that this approach has the potential to differentiate soft candies prepared with different syrups.

## 3. Materials and Methods

### 3.1. Materials

Commercial sucrose (Bal Kupu, Turkey) and bovine (250 bloom value) gelatin were purchased from Sigma-Aldrich Chemical Co. (St. Louis, MO, USA) and corn syrups were kindly provided from Sunar Misir Entegre Tesisleri Sanayi ve Ticaret A.S. (Adana, Turkey). The specific details of the syrups are provided in Table 3.

There are three fructose syrups (SBF10, SHFSLF20, and SMF42) and two glucose syrups (SCG40 and SCG60). The letter ‘F’ at the end of the syrup names refers to fructose syrup whereas ‘G’ refers to glucose syrup. The letters before ‘F’ and ‘G’ are specific product names given by the manufacturer. Then, the numbers in the syrup names represent the fructose content in fructose syrups and glucose content in the glucose syrups. Accordingly, SBF10, SHFSLF20, and SMF42 fructose syrups have fructose contents of 10, 20, and 42%, respectively, as indicated in Table 3. The same rule applies to glucose syrups where SCG40 and SCG60 glucose syrups contain approximately 40 and 60% glucose, respectively. Sodium azide (≥99.99% trace metals basis) (Sigma-Aldrich Chemical Co., St. Louis, MO, USA) was also added at a final concentration of 0.01% (*w*/*w*) in all formulations to prevent microbial growth.

### 3.2. Sample Preparation

Soft candies were prepared by using bovine gelatin along with five different corn syrups and sucrose. Initially, 8 g gelatin was mixed with 15 mL distilled water. Simultaneously, sucrose, corn syrup, and 17 mL distilled water were mixed in another container. The total amount of corn syrup and sucrose was fixed to 60 g. Their exact amounts for each formulation are shown in Table 4. When the mixture temperature reached 100°C, mixing was stopped. Subsequently, these two mixtures were added to a beaker and stirred at 350 rpm, 85 °C. They were mixed until Brix values reached 70 °Bx. Then, the mixture was poured into molds and allowed to cool [1].

Here, soft candies were named according to the corn syrup from which they were produced. A detailed explanation of the syrup names was given in Section 2.1. Additionally, corn syrup concentration of the soft candy samples varied as shown in Table 4. Soft candies contain either 30% corn syrup (fructose or glucose syrup depending on the last letter of the given names) or 60% corn syrup. For instance, SCG40-30% represents the soft candy samples containing 30% SCG40 corn syrup in their formulation while SCG40-60% refers to the presence of 60% of the same syrup (SCG40) in the candy formulation. The same logic applies to all samples.

### 3.3. Water Activity Measurements

Water activity (a_w_) was measured by Aqualab 4TE (Meter Group, Pullman, WA, USA). Samples were cut into thin layers and a_w_ was measured at 25 °C.

### 3.4. Texture Profile Analysis

Hardness values of the soft candies were measured using a Texture Analyzer (Brookfield Ametek CT3, TA18 probe, Middleboro, MA, USA). 0.5 N load cell was used, and test speed was set as 1 mm/s [2].

### 3.5. Brix Measurements

Brix values were measured by using HANNA HI 96801 Refractometer (HANNA Instruments, Leighton Buzzard, UK). 

### 3.6. Time Domain Nuclear Magnetic Resonance (TD-NMR) Relaxometry Experiments

Time Domain (TD)-NMR Relaxometry measurements were conducted by using a 0.5 TD-NMR instrument operating at a ^1^H Larmor frequency of 22.34 MHz (Spin Track, Resonance Systems GmbH, Kirchheim/Teck, Germany). For measurement of spin-lattice relaxation times (T_1_), saturation recovery (SR) sequence was used. Repetition time and time of observation were both set as 300 ms. The number of scans was kept as 8 [2]. Since preliminary transverse relaxation experiments showed no clear T_2_ trend with respect to changing parameters, T_2_ results were not included in this study. SE and MSE experiments were conducted with a repetition time of 10 s and the number of scans was set to 16 [8]. Goldman–Shen sequence was used for Spin Diffusion experiments with a repetition time of 1s and number of scans of 64 [10]. The distance between the first (excitation) and second (flipping of bounded water transverse magnetization to longitudinal direction) pulses in Goldman–Shen sequence was set to 25 µs, assuming that the FID signal of rigid component was effectively decayed. t^0.5^ was determined according to known “initial rate approximation” approach [33]. The initial slope of the polarization transfer curve was fitted with linear function, and the intercept to the x-axis was considered as t^0.5^.

### 3.7. Statistical Analysis

Statistical analysis of the experimental data was performed by Minitab (Minitab Inc., Coventry, UK) software via analysis of variance (ANOVA). Three replicates were performed for each experiment and the mean results were presented with standard deviations for table data and standard errors for the graph data. Comparison of the results were performed by Tukey’s comparison test at 95% confidence interval.

## 4. Conclusions

In this study, we investigated soft candies prepared by five different corn syrups at different syrup concentrations. Some physical properties of the samples such as a_w_, hardness and °Brix were measured and differences were determined. A conventional TD-NMR approach of longitudinal relaxation time was also measured but it was found as an insufficient indicator to differentiate the samples. Later on, SE/MSE sequences were used on the samples in order to calculate the M_2_ values. Crystallinity of the fructose containing samples were found to be the lowest which was in accordance with the basic “food chemistry” facts. Spin diffusion calculations enabled us to calculate the interface thickness and the results showed a good variation within the samples. Presence of fructose affected the thickness significantly and the thickness increased with increasing fructose concentration. Thus, increased interface thickness was associated with lower crystallinity values. That was an important finding, but in order to develop a generalized method for identifying the nature of the syrup in unknown samples, further experiments are required. Physical parameters should be known and by using a large dataset, multivariate analysis models can be constructed, and the thickness values could indicate the type of the syrup used in the formulations. To conclude, in this study, we have shown the fact that there are these ‘non-conventional NMR methods’ used in polymer science literature and they have significant potential to solve food quality-related problems that the industry is facing.

## Figures and Tables

**Figure 1 molecules-27-06745-f001:**
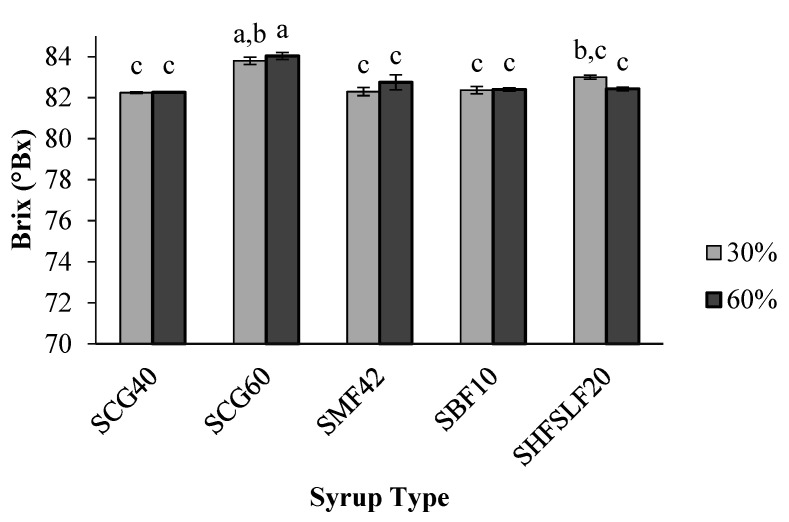
Effect of corn syrup type and concentration on the Brix of the formulated candies. Small letters indicate the significant differences (*p* < 0.05) between the samples. Each bar represents a different sample. Errors are represented as standard errors.

**Figure 2 molecules-27-06745-f002:**
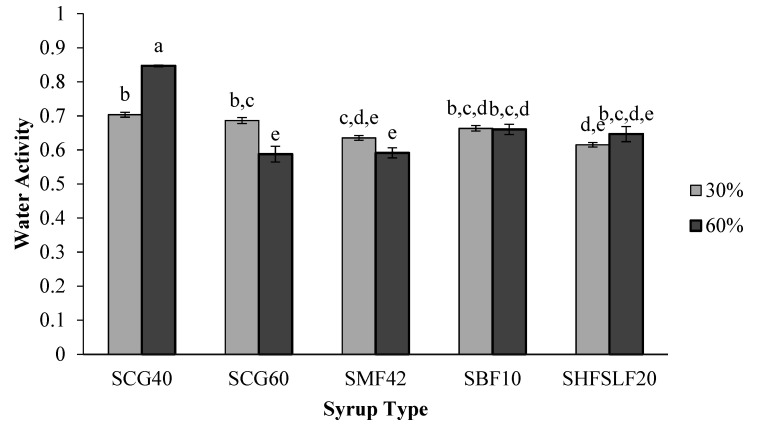
Effect of corn syrup type and concentration on the a_w_ of the formulated candies. Small letters indicate the significant differences (*p* < 0.05) between the samples. Each bar represents a different sample. Errors are represented as standard errors.

**Figure 3 molecules-27-06745-f003:**
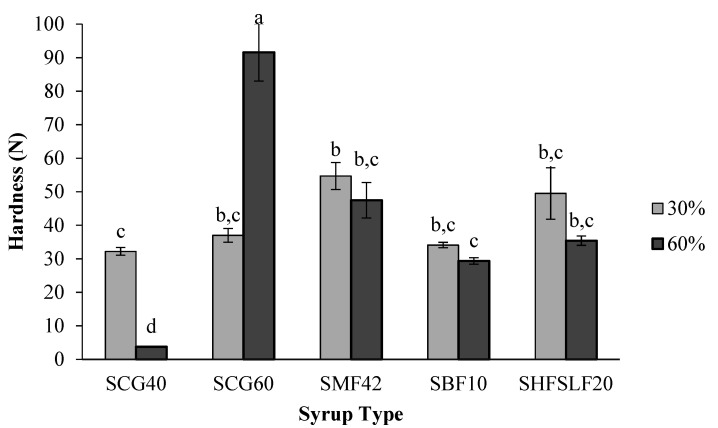
Effect of corn syrup type and concentration on the hardness of the formulated candies. Small letters indicate the significant differences (*p* < 0.05) between the samples. Each bar represents a different sample. Errors are represented as standard errors.

**Figure 4 molecules-27-06745-f004:**
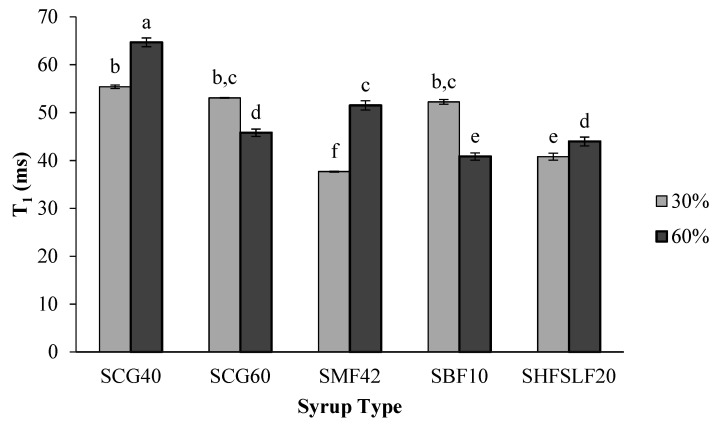
Effect of corn syrup type and concentration on the T_1_ of the formulated candies. Small letters indicate the significant differences (*p* < 0.05) between the samples. Each bar represents a different sample. Errors are represented as standard errors.

**Figure 5 molecules-27-06745-f005:**
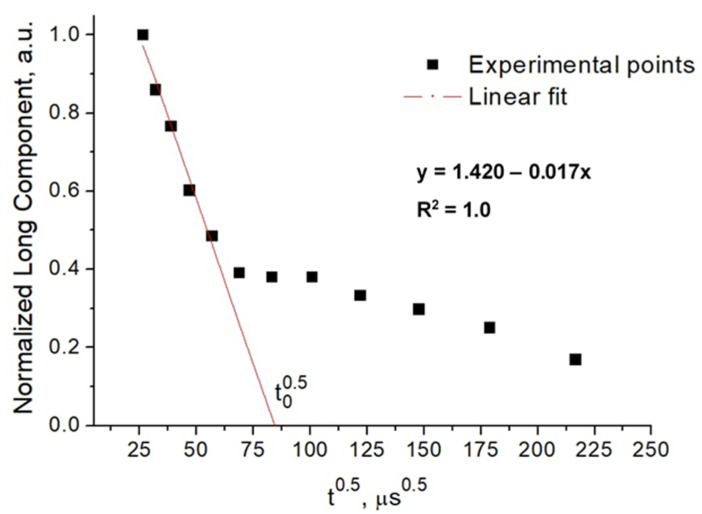
An example of the determination of the parameter t^0.5^ in the Goldman–Shen experiment.

**Table 1 molecules-27-06745-t001:** Second moment (M_2_*10^8^ Hz^2^) of the candy samples.

	M_2_ (SE)		M_2_ (MSE)	
Syrup Amount (%)	30	60	30	60
Syrup Type				
SCG40	7.00 ± 0.10 ^a,b^	7.46 ± 0.07 ^a^	11.76 ± 0.08 ^a,b^	11.99 ± 0.07 ^a^
SCG60	6.86 ± 0.06 ^a,b^	7.25 ± 0.05 ^a,b^	11.21 ± 0.14 ^a,b^	12.02 ± 0.15 ^a^
SMF42	6.14 ± 0.27 ^c^	5.31 ± 0.08 ^d^	9.21 ± 0.23 ^c^	8.55 ± 0.35 ^c^
SBF10	6.77 ± 0.15 ^b,c^	6.75 ± 0.13 ^b,c^	10.92 ± 0.24 ^b^	11.00 ± 0.12 ^b^
SHFSLF20	7.19 ± 0.10 ^a,b^	6.69 ± 0.09 ^b,c^	11.60 ± 0.08 ^a,b^	11.14 ± 0.09 ^a,b^

Small letters indicate the significant differences (*p* < 0.05) between the samples in each column. Errors are represented as standard deviations.

**Table 2 molecules-27-06745-t002:** Interface layer thickness of the soft candies calculated by Goldman–Shen pulse sequence.

Interface layer thickness (Å)		
Syrup Amount (%)	30	60
Syrup Type		
SCG40	47.5 ± 1.8 ^g^	62.2 ± 1.4 ^d,e^
SCG60	50.2 ± 1.5 ^f,g^	69.4 ± 2.4 ^d^
SMF42	83.6 ± 1.8 ^c^	641 ± 3.8 ^a^
SBF10	57.7 ± 2.4 ^e,f^	95.8 ± 1.7 ^c^
SHFSLF20	60 ± 1 ^e^	113 ± 3.5 ^b^

Small letters indicate the significant differences (*p* < 0.05) between the samples in the entire table. Errors are represented as standard deviations.

**Table 3 molecules-27-06745-t003:** Specifications of corn syrups.

Product Name	Brix (°Bx, 20 °C)	Glucose (%)	Fructose (%)
SBF10 Syrup	79 ± 1	36 ± 2	10.5 ± 2.5
SHFSLF20 Syrup	79 ± 1	27.5 ± 2.5	20 ± 3
SMF42 Syrup	70 ± 1	51 ± 3	42.5 ± 2.5
SCG40 Syrup	83 ± 1	40.5 ± 3.5	-
SCG60 Syrup	81.75 ± 1.25	60.5 ± 3.5	-

**Table 4 molecules-27-06745-t004:** Candy formulations.

Sample Name	Sucrose (g/100 g mix)	Glucose (g/100 g mix)	Fructose (g/100 g mix)
SCG40-30%	30	11	-
SCG60-30%	30	17.1	-
SMF42-30%	30	14.4	12
SBF10-30%	30	10.2	2.4
SHFSLF20-30%	30	6.9	5.1
SCG40-60%	-	22	-
SCG60-60%	-	34.2	-
SMF42-60%	-	28.8	24
SBF10-60%	-	20.4	48
SHFSLF20-60%	-	13.8	10.2

## Data Availability

Not applicable.

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
