# Peer review of "Non-Conventional Time Domain (TD)-NMR Approaches for Food Quality: Case of Gelatin-Based Candies as a Model Food"

_molecules, 2022, doi:10.3390/molecules27196745_

Round 1
Reviewer 1 Report
The research is challenging and is based on the use of a non-conventional TD-NMR approach of Solid Echo/Magic Sandwich Echo and Spin Diffusion to investigate the composition of jellies as food model. By the obtaining of jellies, starch hydrolysates (named corn syrups, too) are normally used, together with sugar and a gelling agent. In my opinion, the choice of the starch hydrolysates as models is not very appropriate because the starch hydrolysis products are very complex. Depending on the hydrolysis method, these syrups are very different in composition and not only the glucose content is important, but the percentage of other sugars as maltose, maltotriose and olygosaccharides, too. More, the syrups used in this research are containing fructose, which is obtained from dextrose syrups by inversion.
The writing style is poor. Discussion of the results is quite absent. The authors recognize that a calibration curve for crystallinity measurement should be necessary, but they don’t have any.
The authors should improve substantially their manuscript and some proposals for improvements are given below:
In Introduction:
-The formulation “Gelatin based candies are … composed of … gelling agent such as gelatin or pectin” is not quite correct. Gelatin-based candies are based especially on gelatin; pectin can be added, too, to modify the texture. The “or” in this phrase should be replaced.
In Materials and methods:
-The abbreviations of syrups (table 1) are related with the composition and the content in the major (or the most important) compound and it has to be described in the material;
-The same observation for table 2 as for table 1, to explain the abbreviations;
-The experimental design (the recipes) should be clearer explained. The authors should justify their chose of the ratio between the sugars in the formulations in table 2. It is not at all clear how and why all these syrups were chosen;
-How the glucose and fructose contents were determined, as given in table 2?
- How the syrups containing fructose were obtained?
- The point 2.6. is not indicated. No literature resources are given on the working methodology.
In Results and discussion:
-The phrase at lines 121-123 sounds very strange, it has to be reformulated;
-The phrase at lines 130-132 is very unclear;
- Where is figure 1?
- line 180: relaxations times changed in the range of 3-47 ms, but eh value of 3 ms is absent in the values in figure 2;
- In all cases, by presenting the results related with the Brix values, water activity and hardness, minimal a table could be included, with the results, for clarity. Now, generally, it is very difficult to follow the results;
- No discussion of the results compared with the literature is done, this a weak point of the manuscript, in this form.
Author Response
Comments and Suggestions for Authors
The research is challenging and is based on the use of a non-conventional TD-NMR approach of Solid Echo/Magic Sandwich Echo and Spin Diffusion to investigate the composition of jellies as food model. By the obtaining of jellies, starch hydrolysates (named corn syrups, too) are normally used, together with sugar and a gelling agent. In my opinion, the choice of the starch hydrolysates as models is not very appropriate because the starch hydrolysis products are very complex. Depending on the hydrolysis method, these syrups are very different in composition and not only the glucose content is important, but the percentage of other sugars as maltose, maltotriose and olygosaccharides, too. More, the syrups used in this research are containing fructose, which is obtained from dextrose syrups by inversion.
We thank the reviewer for their comment. We agree that the systems that we have studied are complex and possess different compositions but we wanted to simulate a realistic condition for our samples. Additionally, the main difference in crystallinity is expected to come from the presence and/or absence of sucrose. There is also a reliable literature data on the crystallization, solubility etc. properties of glucose and other higher molecular weight sugar. We hope that the revisions we have made improved the quality of the manuscript.
The writing style is poor. Discussion of the results is quite absent. The authors recognize that a calibration curve for crystallinity measurement should be necessary, but they don’t have any.
We have worked on the manuscript in order to improve the writing style, language and depth of the discussion. We hope that the revisions improved the manuscript. For the calibration curve, we have referred to the study of Grunin et al. (2019). As we have mentioned in the text, our primary aim was to make a comparison between different samples, not to propose quantitative crystallinity values. Thus, the M2 results should be regarded as relative values for comparison. Since the samples differed a lot in their compositions, preparing a calibration curve would not be feasible in this case.
The authors should improve substantially their manuscript and some proposals for improvements are given below:
In Introduction:
-The formulation “Gelatin based candies are … composed of … gelling agent such as gelatin or pectin” is not quite correct. Gelatin-based candies are based especially on gelatin; pectin can be added, too, to modify the texture. The “or” in this phrase should be replaced.
We have revised the sentence as suggested.
In Materials and methods:
-The abbreviations of syrups (table 1) are related with the composition and the content in the major (or the most important) compound and it has to be described in the material;
We have given detailed explanations about the syrup types in the ‘Materials’ section (section 2.1.) in the revised form of the manuscript.
-The same observation for table 2 as for table 1, to explain the abbreviations;
We have also provided detailed descriptions of the soft candy names in the related section (section 2.2.)
-The experimental design (the recipes) should be clearer explained. The authors should justify their chose of the ratio between the sugars in the formulations in table 2. It is not at all clear how and why all these syrups were chosen;
In this study, our primary aim was to show that the changes in the physicochemical properties of the samples induced by corn syrups could be analyzed by non-conventional TD-NMR parameters. We believe that both the concentration and type of the corn syrup used in soft candy formulations induce differences that could be tracked by some NMR parameters. Since fructose to glucose ratio in the corn syrups is expected to reveal different results, firstly we have used five different corn syrups with varying fructose-glucose contents. However, these corn syrups are provided from the manufacturer so that we could not practically adjust the composition of the syrups. Additionally, two corn syrup concentration levels (30% & 60%) were used to compare the effects of corn syrup concentration in the soft candies. Our final concern was the presence of sucrose in the candy formulations since industrially produced soft candies also contain some amount of sucrose in their formulations. For this purpose, we have fixed the total corn syrup + sucrose content to 60% and replaced the 30% of corn syrup with 30% sucrose in half of the samples. In this way, we were able to investigate the effects of both the changing corn syrup concentration and presence/absence of sucrose in the candy formulations. The recipes may look complicated but there were many correlations (trends) with respect to the changes in the candy recipes and we have revised the manuscript to make our points clearer.
-How the glucose and fructose contents were determined, as given in table 2?
We have provided different types of corn syrups with varying glucose and fructose contents from the manufacturer. The contents of the syrups were provided by the company. These are specific products produced by the company. Since we know the syrup compositions-concentrations used in the soft candy formulations and we also know the concentration of the other ingredients (sucrose, water…), we have calculated the fructose and glucose contents in the final soft candy samples. So, we did not directly measure the final glucose and fructose contents in the soft candies.
- How the syrups containing fructose were obtained?
The fructose-glucose compositions of the corn syrups were specified by the manufacturer. So, we did not produce fructose containing syrups ourselves.
- The point 2.6. is not indicated. No literature resources are given on the working methodology.
We have included the point 2.6. We have also included the literature resources on the working methodology into the related sections, as suggested.
In Results and discussion:
-The phrase at lines 121-123 sounds very strange, it has to be reformulated;
We have revised these sentences as suggested.
-The phrase at lines 130-132 is very unclear;
We have revised these lines as suggested.
- Where is figure 1?
There was an error in the previous form of the manuscript in terms of number of figures used. We have revised the manuscript accordingly and added new figures. The numbering of the figures was corrected in the revised form.
- line 180: relaxations times changed in the range of 3-47 ms, but eh value of 3 ms is absent in the values in figure 2;
We have removed the T2 figure form the manuscript, thus this should no longer be a problem.
- In all cases, by presenting the results related with the Brix values, water activity and hardness, minimal a table could be included, with the results, for clarity. Now, generally, it is very difficult to follow the results;
We have included new figures (water activity, Brix and hardness figures) in the revised form of the manuscript. Moreover, we have divided the discussion part into subsections. In this way, we believe it became easier to follow the results. Since the number of the samples and the measured parameters is high, preparation of a minimal table was not possible.
- No discussion of the results compared with the literature is done, this a weak point of the manuscript, in this form.
We also have realized that the depth of the discussion was inadequate and the literature comparison was poor in our manuscript. We have made an extensive revision in the ‘results & discussion’ section. We also have provided more related references in order to compare-discuss our results. However, we must also state that the related literature on this topic is quite limited. This is a newly introduced topic into the food science literature and we have tried to provide as much literature information as we can.
Reviewer 2 Report
In this manuscript, the authors utilized SE, MSE and SD NMR methods to analyze the soft candies. In comparison the conventional TD-NMR (T1 and T2) measurements, these parameters are more sensitive indicators for differentiating the food samples. It is an interesting paper, I recommend its publication on Molecules, but with minor revisions.
1. The SE, MSE and SD NMR methods have been successfully used for investigating polymer materials. The related literatures should be cited properly.
2. The connection between the obtained NMR parameters and physical properties is lacking, and it’s suggested to include the corresponding discussions in the manuscript.
Author Response
Comments and Suggestions for Authors
In this manuscript, the authors utilized SE, MSE and SD NMR methods to analyze the soft candies. In comparison the conventional TD-NMR (T1 and T2) measurements, these parameters are more sensitive indicators for differentiating the food samples. It is an interesting paper, I recommend its publication on Molecules, but with minor revisions.
We thank the reviewer for their comment. We have revised the manuscript and we hope that these revisions improved the manuscript.
- The SE, MSE and SD NMR methods have been successfully used for investigating polymer materials. The related literatures should be cited properly.
We have included more literature information and cited the related references in the revised form of the manuscript, as suggested.
- The connection between the obtained NMR parameters and physical properties is lacking, and it’s suggested to include the corresponding discussions in the manuscript.
We have included the suggested discussions in the manuscript and revised the whole ‘results & discussion’ section.
Reviewer 3 Report
My comment is as follows.
1. It is difficult to understand table 1. Does the Brix value mean the total percentages of sugars in the samples? On page 4, you mention that SCG40 and SCG60 contains maltose. Can you include them in table 1? The sum of Glucose and Fructose is much less than the Brix except SMF42. Does it mean there are unknown sugars in the samples?
2. The data of water activity and hardness are not shown. Please add tables.
3. Please show the 1D (MSE) spectra of ten samples in the supplementary information. Are there three components as you showed in Eq. (1). Please mention about their assignments.
4. The 2nd moment data in table 2 need more interpretation. You defined M2^{cr} and M2^{am} in Eq. (2), but there is only a single value in table 2. The unit is lacking in table 2. Please compare these with the values of sucrose crystals and dried gelatin samples.
5. The description of the Godman-Shen experiment is very poor. In the method section the time delay between 1st and 2nd pulses is not mentioned. According to Cheung and Gerstein (J. Appl. Phys. 52, 5517 (1981). ), M2 in Eq. (4) must be replaced by D. It will make the unit in the both sides of Eq. (4) consistent. The author did not define t0 and it is not clear how it was determined. What is the amplitude of Figure 4? The paper by Cheung and Gerstein described their procedure very clearly.
6. The model of the domain structure is not clear. What is the rigid (or soft) components?
Author Response
Comments and Suggestions for Authors
My comment is as follows.
- It is difficult to understand table 1. Does the Brix value mean the total percentages of sugars in the samples? On page 4, you mention that SCG40 and SCG60 contains maltose. Can you include them in table 1? The sum of Glucose and Fructose is much less than the Brix except SMF42. Does it mean there are unknown sugars in the samples?
Brix value refers to the total percentages of sugars in the samples. There should also be higher molecular weight sugars in the samples (maltose, some oligosaccharides etc.) introduced into the system by corn syrups since Brix values are larger than the total glucose – fructose concentration. However, we have provided these corn syrups directly from the manufacturer and these were not determined by the provider. We have mentioned about the possible effects of higher molecular weight sugars in the syrups and made some revisions in the manuscript.
- The data of water activity and hardness are not shown. Please add tables.
We have added the figures of water activity, Brix and hardness values in the revised form of the manuscript.
- Please show the 1D (MSE) spectra of ten samples in the supplementary information. Are there three components as you showed in Eq. (1). Please mention about their assignments.
We have calculated the M2 as described in the paper of Grunin et al (2019). Second moment values of MSE data were used for the ‘spin diffusion calculation’. We realized that we have made a mistake with putting the 3-component Abrahamian equation since M2 was not calculated by using this classical approach. That is why a 3-component assignment was not made. Sorry for the confusion we caused.
- The 2nd moment data in table 2 need more interpretation. You defined M2^{cr} and M2^{am} in Eq. (2), but there is only a single value in table 2. The unit is lacking in table 2. Please compare these with the values of sucrose crystals and dried gelatin samples.
Unit was added. The model was removed. Sorry for the confusion we caused.
- The description of the Godman-Shen experiment is very poor. In the method section the time delay between 1st and 2nd pulses is not mentioned. According to Cheung and Gerstein (J. Appl. Phys. 52, 5517 (1981) ), M2 in Eq. (4) must be replaced by D. It will make the unit in the both sides of Eq. (4) consistent. The author did not define t0 and it is not clear how it was determined. What is the amplitude of Figure 4? The paper by Cheung and Gerstein described their procedure very clearly.
Firstly, we have corrected the error in Eq. (4) and replaced M2 with Dsd. Secondly, we have added the requested descriptions of the Godman-Shen experiment in the Materials & Methods section (section 2.6).
- The model of the domain structure is not clear. What is the rigid (or soft) components?
It is really difficult to visualize the domain structure in candy. But what we have thought was the presence of a semi-crystalline matrix where there is the gel; sugar crystals and water in the candies. That is why rather than defining a real structure ‘a detailed description’ was not provided. An explanation was added to the text to make it easier for the readers.

Round 2
Reviewer 1 Report
The authors improved significantly their manuscript and they were receptive to all my observations, by adding explanations or modifying their text.
Author Response
Thank you for the feedback :)
Reviewer 3 Report
I think that the TD-NMR is advantageous because it is less-expensive and therefore more democratic and educational. It might help to redistribute science. In that sense, your paper should be more reader-friendly and more informative.
The description of the sec. 3.1-3.3 are greatly improved compared to the former version. However, I still feel that se. 3.6 is problematic. Two equations are given with a few references, so the readers or students may feel difficult to follow. The followings are my final suggestions. If you agree, please add the proper references and change the paper.
1. Eq. (1) was derived by reference 29 and compared with the theoretical results by Lowe and Gade for CaF2. The spin diffusion in CaF2 was measured in 1998 by Zhang and Cory and was compared with theory in
D. Greenbaum et al., Phys Rev. B71, 054403 (2005).
The papers by the Cory's group listed the references written by physicists before ref. 29.
2. Eq. (2) was derived by Cheung and Gerstein in
J. Appl. Phys. 59, 5517-5528 (1981),
by solving the diffusion equation. Eq. (11a) in this paper corresponds to Eq. (2) of your paper. They derived the t^{0.5} dependence and pointed that this behavior was observed in the kinetic experiments by citing ref. 24 and 25. Schmidt-Rohr might read these references and showed that erfc(x/(4Dt)^{0.5}) is the solution of diffusion equation.
Acta Polymer., 44, 1-17 (1993).
The function erfc may be found in J. Crank "The Mathematics of Diffusion" or Arfken and Weber "Mathematical Methods for Physicists" chap 10.
3. Figure 5 is still insufficient. Does "Amplitude" mean "the long component of FID"? Is the fitting function C0+C1*(1-(t/t0)^{0.5})? How did you determine C0? Please write the fitting function instead of t0^{0.5}. Fig. 5 will be understood that t0 was determined from the intersection between the dashed line and Amplitude=0.175. It was very mysterious to me.
4. In sec. 3.6 the interpretations of T1 depend whether the system belongs to higher or lower temperature side of the T1 minimum. The T1 values can be also compared with the reported T1 minimum value of pure samples. T1row measurements will provide the information on the frequency dependence.
Author Response
They are responded in the attached document.
